# Analyzing the Traffic Operational Performance of School Pick-Up and Drop-Off Dynamics in Saudi Arabia

Sherif Shokry [1,*], Ali Alrashidi [1] and Marwa Elbany [2]

1    The Center of Road Traffic Safety, Naif Arab University for Security Sciences, Riyadh 11452, Saudi Arabia; aalrashidi@nauss.edu.sa
2    Transportation and Traffic Engineering Department, Faculty of Engineering, Port Said University, Port Said 42526, Egypt; mr_elbany@eng.psu.edu.eg
*    Correspondence: sshokry@nauss.edu.sa

**Abstract:** In seeking sustainable, safe, and efficient school commuting tours as non-recurring sources of congestion, it is essential to investigate the dynamic interaction between school students' pick-up and drop-off (P&D) movements and the traffic operational performance of the surrounding area. This study investigates the traffic operational performance in the vicinity of schools at various P&D time intervals. The Travel and Planning Time indices (TTI, PTI), along with the Level of Service (LOS) are utilized as traffic operational performance indices in this article. A Python script was developed to employ the Google API for estimating the travel times from a real traffic dataset comprising 40 schools distributed across six cities in the Kingdom of Saudi Arabia (KSA). The results indicate that LOS varies from C to D for all cities except Riyadh, which exhibits the poorest traffic performance during P&D time intervals. This paper serves as a guideline for city planners and policymakers seeking to provide valuable insights to enhance traffic operational performance in Saudi Arabia.

**Keywords:** school vicinity; operational performance indices; level of service (LOS); travel time index (TTI); capacity analysis; Google API

## 1. Introduction

The dynamic interaction between school students' P&D movements and traffic operational performance has become a focal point for researchers, urban planners, and policymakers. The nexus demands a comprehensive understanding to address its multifaced impacts and devise effective strategies for enhancing traffic safety in this critical context. Traffic congestion stands out as a pervasive challenge across the globe, not exempting major Saudi Arabian cities. The rapid urbanization and economic growth experienced across these cities has resulted in substantial private vehicle ownership growth rates, outpacing the expansion of road networks. Consequently, the streets in Saudi cities frequently grapple with severe congestion during peak hours, which leads to significant delays, undoubtedly negative frustration, and substantial environmental pollution. In contemporary urban societies, school commutes have emerged as a significant contributor to the decline in traffic operational performance and safety indicators on road networks [1].

The inadequacy of public transportation services plays a significant role in congestion levels throughout these cities [2]. Despite the significant improvements in the transportation system in Saudi cities over the last two decades, it remains insufficient to accommodate the high traffic demand. Consequently, numerous urban areas in Saudi Arabia encounter accessibility challenges, compelling residents to rely heavily on their private vehicles for their daily commutes.

Numerous studies have addressed the complexity of modeling congestion in school neighborhoods, highlighting the necessity for specialized expertise and software to effectively model congestion around schools. This necessity stems from the intricate interplay between transportation, land use, and behavioral factors that impact traffic congestion

in this sensitive area. The transportation system, land use, and road users' behavior can be intertwined in multifaced ways, exerting an influence on traffic congestion around schools. This underscores the pressing need to consider a diverse array of data sources and modeling techniques to accurately assess and mitigate traffic congestion challenges in this critical domain.

This article serves to analyze the traffic operational performance in the vicinity of schools utilizing the TTIs by calculating the travel time between a certain origin and destination at each period/the same road segments free flow conditions travel time in the three main selected time intervals, namely drop-off (morning time) and pick-up (afternoon time) and off-peak periods outside of those intervals. By grasping the intricacies of pick-up and drop-off traffic patterns, policymakers can draw strategies to mitigate congestion and enhance safety conditions in school zones. This proactive approach fosters the development of communities that are both sustainable and conducive to raising the quality of life and safeguarding welfare.

This paper is organized into the following sections: Section 2 provides the state-of-the-art related to the research topic. Meanwhile, the theoretical capacity analysis to calculate mobility indices such as LOS (i.e., PTI and BTI) and TTI is presented in Section 3, the materials and methods and section. The materials and methods section also presents the research methodology; the research objectives are presented in addition to the data collection circumstances of the real-life traffic datasets used in this study. The output of the analysis and the main findings and their implications are discussed in Section 5. Finally, Section 6 offers a conclusive synthesis of the study outcomes and their far-reaching implications.

## 2. Literature Review

To model traffic performance, some researchers have focused on the behavioral aspect, emphasizing the significant burden placed on families as a result of the inadequacy of public transportation services. This burden means that daily commutes must be arranged for both parents and kids, thus exacerbating the growing reliance on private transportation [1,3]. Their findings concluded that suitable schools are often not located near these families' residences, rendering them inaccessible on foot or by bicycle due to the considerable distances that must be travelled. Consequently, parents are forced to drive their children to school to ensure their safety, albeit at the expense of exacerbating traffic congestion and discouraging active models of mobility [4,5].

Recently, the phenomenon of driving children to school in their parents' private vehicles emerged as a pressing concern. Consequently, there has been a rapid and substantial surge in vehicle ownership among families, resulting in heightened congestion on roads leading to schools during peak hours [6,7]. This traffic congestion exacerbates issues associated with road traffic and imposes adverse environmental impacts, notably through heightened greenhouse gas emissions [8]. A previous study explored the dynamic interaction between school students' P&D routines and the traffic operational performance, igniting a heightened interest among researchers, urban planners and policymakers to comprehend and mitigate the diverse impacts of this daily occurrence through appropriate interventions. During P&D periods, parental vehicles typically converge in the vicinity of the school, precipitating traffic disruption, double parking, and intersections closures, particularly in the absence of designated parking lots and access points to school bus loading areas and other related facilities [9].

The operational performance near schools is significantly compromised during these times, resulting in congestion as roads' capacities are exceeded. This can significantly degrade the LOS of the road network during morning and afternoon peak hours [10]. Travel times are prolonged for various road users, including school buses, emergency vehicles, and residents. Haphazard and unlawful parking practices lead to lane closures and reduced road capacity, while impatient drivers resort to risky maneuvers to circumvent queues, consequently elevating the risk of traffic accidents. Gachanja J. introduced this disruptive impact on traffic conditions around schools transcends mere inconvenience,

encompassing a spectrum of environmental and socioeconomic ramifications. In the pursuit of a sustainable, safe, and efficient transportation system, it is imperative to discern the precise factors involved in students' P&D activities that contribute to traffic congestion [11].

Travel time reliability (TTR) serves as a crucial performance metric for monitoring and evaluating traffic performance. Studies examining the TTR performance index have demonstrated that this index, along with TTI and the Buffer Time Index (BTI), exhibits an increase with a higher percentage of variance, commonly known statistically as the coefficient of variation. This underscores the superiority of TTR indicators, which offer tangible benefits to road users. While some of these studies utilized normal distributions and Weibull distributions to ascertain TTR, the inability to incorporate real-time traffic data may compromise the accuracy of reliability predictions. Pu W. estimated BI based on the median, rather than the mean, to mitigate the occurrence of low TTRs, particularly for highly right-skewed travel time distributions. Similarly, a modified BI was introduced as a proposed measure for TTR, employing median values rather than mean travel time, thereby influencing LOS [12].

An extensive analysis of TTR measures was conducted for congestion across the entire Portland, Oregon metropolitan area, utilizing data from 2004 to 2007 encompassing both highways and corridors sections. Yamazaki F. et al. advocated for incorporating reliability metrics alongside traditional congestion metrics to facilitate more informed planning decisions. Another attempt leveraged electronic toll collection data to identify traffic conditions affecting travel time, enabling the evaluation of LOS based on TTR [13]. Meanwhile, Silvano L. B. et al. highlighted the influences of changes in Posted Speed Limits (PSLs) on urban networks' free-flow speeds. The authors investigated the predominant influence of geometric characteristics and functional categories on the mean speed over PSL variations [14]. In a different approach, Al Ghanim J. et al. utilized travel time data to assess the major roads' operational performance during the evening congestion in the southern part of Najaf city. The statistical analysis revealed instances where sectors exhibited high LOS using the V/C ratio criterion practiced but demonstrated low LOS upon observations or when considering TTR as a performance index [15].

## 2.1. Non-Recurring Congestion Sources

The Highway Capacity Manual (HCM) is widely used as a prediction methos for assessing the impacts of various non-recurring congestion sources (i.e., unusual weather conditions, sudden incidents) on operational performance. A meaningful descriptive Congestion Performance Measure (CPM) should exhibit statistical efficiency and reliability. Recurrent congestion was defined as a daily or regular phenomenon during peak hours, resulting in significant delays. The reproducible congestion remains consistent during similar peaks. On the other hand, non-recurrent congestions, as described in the literature, encompass unexpected increases in travel time rates, queue formulation, and mobility reduction as a result of random incidents [16]. In this context, the P&D activities are regarded as non-recurrent congestion.

There are two approaches to categorizing the definitions of congestion: the bottleneck approach and the travel time approach. When the traffic demand surpasses the bottleneck capacity, leading to queue formation which results in upstream interception affecting the whole network, the bottleneck approach proves valuable [17]. Elefteriadou G. R. et al. highlighted that real-world measurements emphasize that capacity is not often an upper bound for LOS E, particularly, when the maximum flow occurs as a result of the congestion occurrence (i.e., breakdown) [18]. In contrast, the travel time approach focused on by Forkenbrock W., which emphasizes economic effects, has demonstrated that the travel time indices provide the most effective means for the prediction of the economical drawbacks of congestion. This approach focuses on commuting speed or trip travel time (travel time between two points) [19].

### 2.2. Google Application Programming Interface (API) Real-Time Traffic Analyzer

The Sydney-based company "Where 2 Technologies" initially developed Google Maps as desktop C++ language. In October 2004, Google acquired the company and transformed the software into a web application. Subsequently, Google purchased a real-time traffic analyzer and a specialized geospatial data visualization company, enhancing the capacities of Maps. In February 2005, the revamped Google Maps was unveiled. Its front end utilizes XML, Ajax, and JavaScript. In June 2005, the Google Maps API was introduced [20]. This API facilitates third-party map integration and servers as a global business locator and other related organizations. Early versions of Google Maps primarily relied on data from traffic sensors, which were predominantly installed by for-profit companies or governmental transportation agencies specialized in traffic data collection [21]. These sensors, employing radar, active infrared, or laser technologies, can determine the passing vehicles' size and speeds. Subsequently, this information is wirelessly transmitted to server hubs [22]. Real-time traffic updates are generated, with the collected data forming part of a historical pool that aids in predicting future traffic patterns.

Now known as the Google Maps Platform, Google Maps API (Application Programming Interface) encompasses approximately 17 different APIs grouped into the following categories: Routes, Places, and Maps. Initially introduced in 2005, the Google Maps API enabled web developers to seamlessly integrate Google Maps into their applications [23,24]. To date, no prior research has investigated the use of TTI during the P&D times as a factor influencing delay and cause congestion. In line with the HCM approach, this study delves into the traffic operational performance in school neighborhoods across various P&D time intervals.

## 3. Materials and Methods

### 3.1. Theoretical Capacity Analysis

Capacity analysis is a recommended methodology for evaluating road networks' or transportation network's abilities to accommodate traffic demand. It involves estimating the theoretical capacity of roads by utilizing various factors such as speed limits, lane width, number of lanes, and road type. Additionally, capacity analysis can be employed to calculate mobility indices such as LOS or TTI based on traffic volume, road capacity, and other performance metrics by estimating the travel time between a certain origin and destination at each period divided by the same road segments free flow conditions travel time as shown in Equation (1) [25]. This equation serves as a valuable factor in traffic congestion and TTR measures. It indicates the mean additional time needed for the peak-hour trip as compared to low traffic or free-flow conditions. The HCM method offers a more intricate and time-intensive approach, providing a comprehensive assessment of LOS based on a wide array of factors [26]. In contrast, the TTI method is simpler and easier to calculate but provides a limited assessment of the TTR. The choice between methods typically hinges on the specific needs and objectives for assessing mobility.

$$TTI = \frac{T_p}{TT_{free-flow}} ,$$ (1)

where

$T_p$: mean travel time during the congested periods;
$TT_{free-flow}$: Travel time during free-flow conditions.

In this study, TTI serves as the chosen traffic performance indicator for measuring congestion levels and delays during P&D times as non-recurring events. The decision to use speed in free-flow conditions instead of the recommended counterparts is based on its better reflection of possible speeds [16]. Additionally, to indicate the LOS thresholds, Buffer Time Index (BTI) and PTI are estimated as shown in Equations (2) and (3), respectively. The 95th% travel time ($TT_{95}$) is divided by the free-flow conditions travel time ($TT_{free-flow}$) to estimate the PTIs [27,28]. On the other hand, BTI is calculated by knowing 95th% travel time

($TT_{95}$) and the estimated mean travel time ($TT_{mean}$) [27,29] as illustrated in the following equations:

$$PTI = \frac{TT_{95}}{TT_{free-flow}}, \tag{2}$$

$$BTI = \frac{TT_{95} - TT_{mean}}{TT_{mean}} \times 100, \tag{3}$$

where

$TT_{95}$:    the 95th% travel time;
$TT_{mean}$:  estimated mean travel time.

Calculating travel time indices for non-recurring movements, such as school pick-up and drop-off periods, special events, accidents, etc., holds significant importance. Identifying areas with high planning indices allows authorities to prioritize improvements or interventions aimed at reducing delays and congestion during non-recurring events. High travel time indices may serve as indicators of areas where emergency response times could be compromised, prompting authorities to implement proactive measures to mitigate potential delays. The method used in this study is presented in the following section.

### 3.2. Research Method

The methodology of this study is designed to be usable in real-life scenarios, leveraging a realistic dataset that encompasses representative factors influencing traffic performance in the schools' vicinities. To fulfill the study objectives, the research methodology is divided into two parts: first, capacity analysis according to delay time and speed data, and second, estimating TTIs and their correlation with the PSL and thresholds of LOS as indicators of traffic operational conditions as shown in Figure 1. The specific objectives of this study were derived to carry out the following:

- Develop a Python script for data collection to detect the same road segments travel time at each time interval in the vicinity of the studied schools.
- Analyze the capacity by calculating TTIs based on the actual travel time and the same road segments free flow conditions travel time in the three selected time intervals: drop-off (morning time) and pick-up (afternoon).
- Highlight the impact of P&D movements on TTIs and differential speed representing delays.
- Estimate TTIs, PTIs, and BTIs of each selected link in the vicinity of the studied schools.
- Pinpoint the relationship between TTIs, BTIs, and PTIs.
- Identify LOS thresholds using average TTIs.
- Estimate LOS at P&D time intervals for each city in the three selected time intervals.

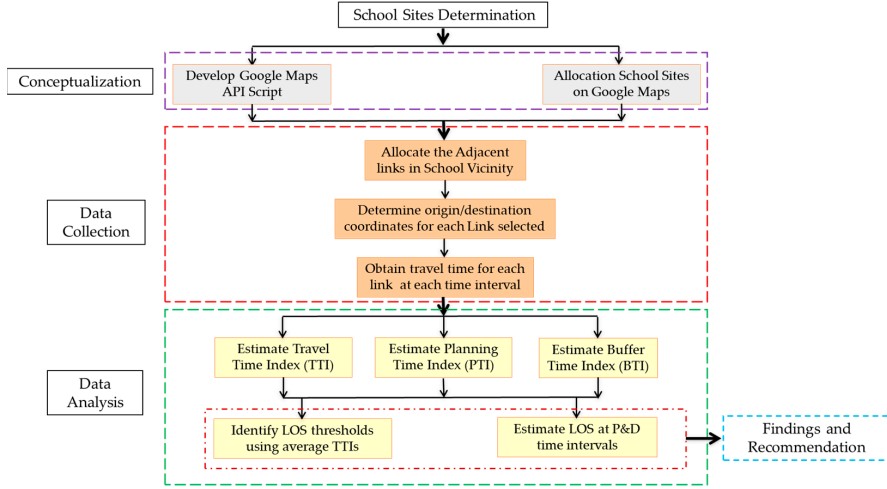

**Figure 1.** The methodological framework.

The algorithm used in this article was designed based on the Google API's given features. First, a request to recall the data needed from the servers is presented. The code output data needed (i.e., distance, avg. duration travel time…etc.) are identified. The coordinates of the start point (origin) and the endpoint (destination) in each direction of the assigned link in the school vicinity are determined. The travel distance is calculated, and the travel time in the off-peak period for each assigned direction is estimated accordingly. At the certain time interval that represents the pick-up and drop-off periods, the code is turned on, and the travel time for each direction on the assigned link is estimated for such a given interval. Finally, the data are saved and exported to spreadsheets for screening, processing, and analysis.

*3.3. Data Collection*

In this research, real-life traffic datasets of the main links around 40 schools in six different cities across KSA were estimated using Google Maps API. For this purpose, a Python script was developed to retrieve the necessary data from the vicinity of the selected schools via Google API for the TTI estimation. Additionally, road geometry data (including road classification, number of lanes per approach, existing medians, and on-street parking property) and traffic directional conditions per approach were obtained based on field observations conducted during 2023 for a total of 242 road segments around 40 schools in six cities representing different regions of KSA, namely Hassa, Makkah, Ha'il, Najran, Abha, and Riyadh, which were collected and analyzed as shown in Figure 2. Aiming to establish the validity and reliability of the findings, the school determination followed a certain criterion. All schools were selected in the Center Business District (CBD) where medium-to-high traffic volumes influence the operational performance significantly. Furthermore, the school vicinity is near the commercial activities and surrounded by different road classified types (i.e., collector, local, minor, main, and arterials) as illustrated in Table 1. The sample involved both boys' and girls' schools as combined sex schools are prohibited by law across the KSA. Moreover, the selected schools are composed of single sites where a separate building exists (i.e., elementary, intermediate, and high school) in addition to the complex school buildings where educational levels are combined (i.e., kindergarten and elementary school building, intermediate and high school building). The schools selected are categorized as having between 200 and 1107 minimum and maximum total number of students, respectively, with an average of 565 students. Observations were conducted over the period between 2023 and 2024, comprising 20 time intervals every 5 min: 10 morning drop-off peak intervals and 10 afternoon pick-up peak intervals. It is worth mentioning that school start time varies from one city to another depending on weather conditions. For instance, in Ha'il, located in the northern region of the Kingdom where weather conditions are colder, school begins at 8:00 AM in summer meanwhile it begins at 9:00 AM in winter, whereas in Riyadh it starts at 6:30 AM and 7:15 AM, respectively.

On the other hand, in regions with higher temperatures, schools typically commerce earlier than their counterparts in cooler areas. Accordingly, drop-off and pick-up peak time intervals differ across cities. Therefore, attention was paid to this factor during the data collection and analysis. Hence, to maintain the validity and reliability of the findings and to avoid data uncertainty, all observations were conducted in normal and clear weather conditions. The studied links are classified based on their operational conditions and posted running speeds into certain categories: collector, local, minor, main, and arterial are compiled in in Table 1. Out of 242 road segments analyzed, 15 are classified as roads, while 28, 64, 95, and 40 are categorized as collector, local, minor, and main roads, respectively.

The posted speed in each link was reported and categorized into different groups (i.e., <40, 40–55, >55 mph) is illustrated in Table 2. The outputs are also clearly described and summarized in the analysis section.

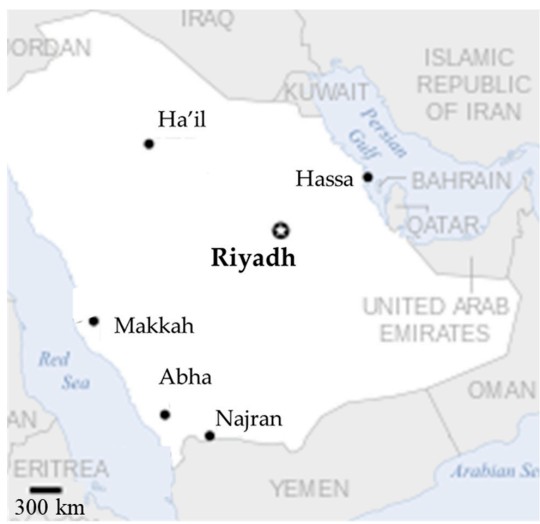

**Figure 2.** A map shows the studied cities.

**Table 1.** The geometric characteristics of studied links.

| Road Category | Total Number | Directional Flow (No. Lanes/Direction) | | | | Existing Median | | On-Street Parking | |
|---|---|---|---|---|---|---|---|---|---|
| | | 1 Lane | 2 Lanes | 3 Lanes | 4 Lanes | Exs. | Not Exs. | Exs. | Not Exs. |
| Arterial | 15 | 0 | 10 | 4 | 1 | 0 | 15 | 1 | 14 |
| Main | 40 | 0 | 14 | 20 | 6 | 7 | 33 | 10 | 30 |
| Minor | 95 | 7 | 63 | 25 | 0 | 16 | 79 | 68 | 27 |
| Collector | 28 | 26 | 2 | 0 | 0 | 28 | 0 | 23 | 5 |
| Local | 55 | 25 | 34 | 5 | 0 | 26 | 38 | 52 | 12 |

**Table 2.** The categorized number of links with the posted speed.

| The Posted Speed (mph) | Total Number of Links |
|---|---|
| <40 | 28 |
| 40–55 | 159 |
| >55 | 55 |

## 4. Analysis and Results

In this part, the interrelationships between the studied performance indices (i.e., TTIs, PTIs, PTIs) with the differential and the posted speeds are analyzed for different P&D intervals. Meanwhile, the LOS at the mean travel time thresholds are determined and discussed accordingly.

### 4.1. TTIs and Differential Speeds

To explain the impacts of P&D movements on the TTIs, which serve as indicators of traffic performance, the TTIs are calculated as illustrated earlier for each link during P&D time intervals every 5 min, encompassing both entry to and exit from schools. The linear regression outputs, representing the best-fit trend line model, are depicted in Figures 3 and 4. Each selected drop-off or pick-up time interval comprises three stages: the beginning of the movement, the running movement, and the end of the movement. During the drop-off (morning) time interval, TTI-1 and TTI-2 were calculated at the onset of the movement, while (TTI-3: TTI-6) were calculated for the concurrent P&D movements. On the other hand, (TTI-7: TTI-9) and TTI-10 were calculated at the conclusion of movement. The interrelationships between TTIs and differential speeds during the drop-off intervals are

illustrated in Figure 3. Likewise, for pick-up (afternoon) time intervals, TTI-11 and TTI-12 were calculated at the onset of the movement. (TTI-13: TTI-16) and TTI-17 were calculated during the P&D movement. TTI-18, TTI-19, and TTI-20 were calculated at the conclusion of the movement. The interrelationships between the TTIs and differential speeds in pick-up intervals are illustrated in Figure 4.

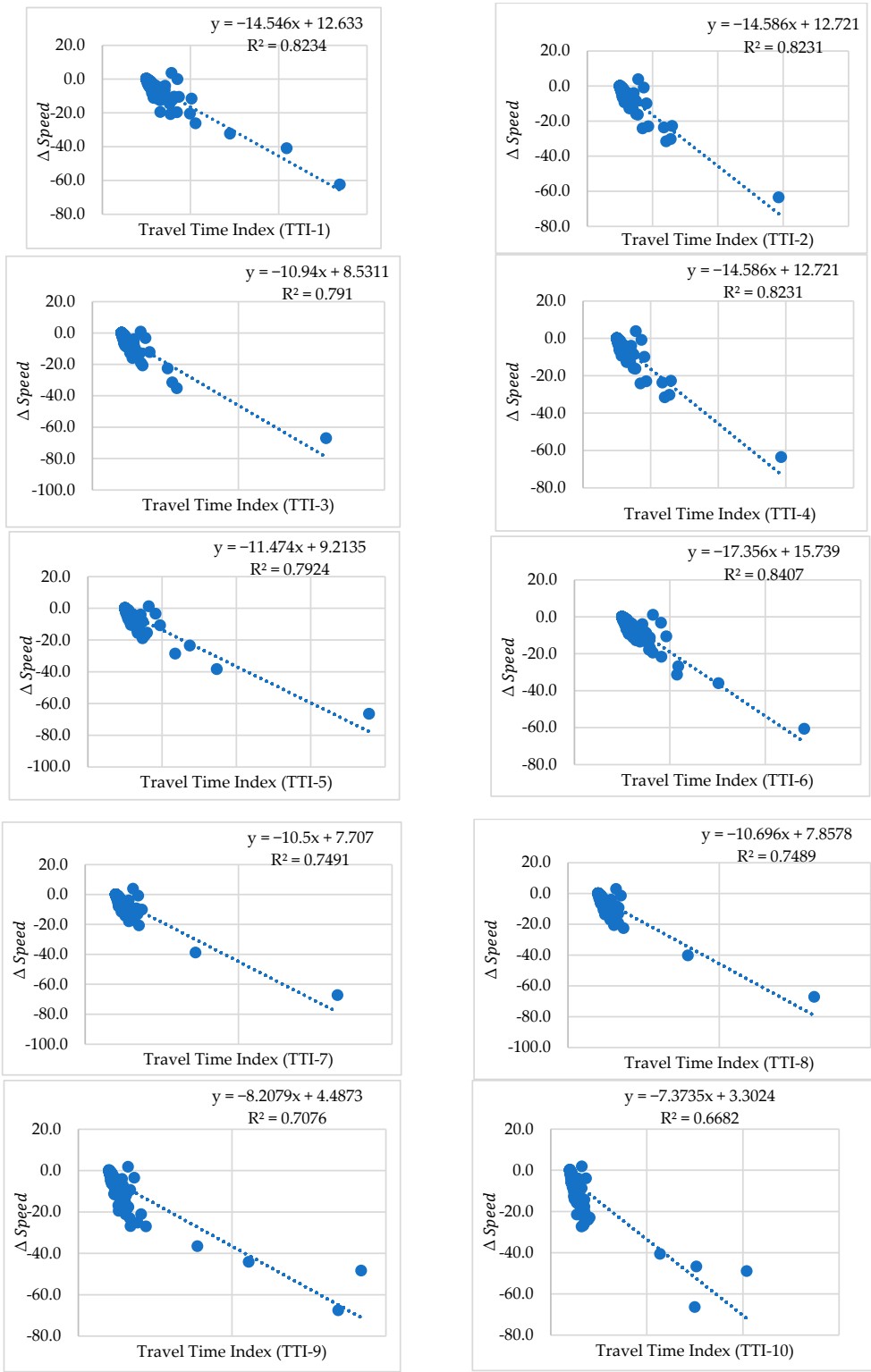

**Figure 3.** Linear correlation between TTIs and average speeds in the morning peak intervals.

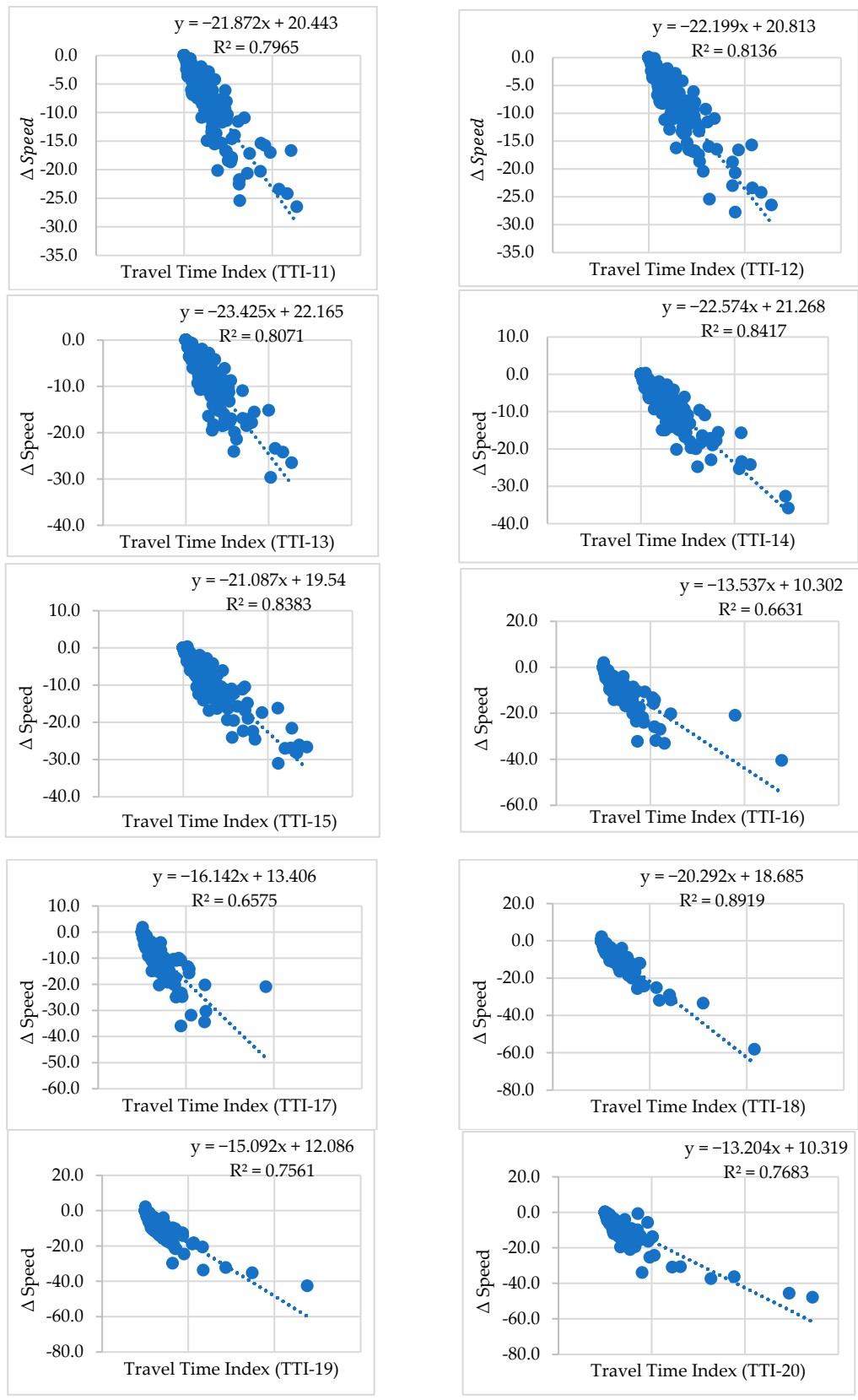

**Figure 4.** Linear correlation between TTIs and differential speeds in the afternoon peak intervals.

The coefficient of determination ($R^2$) indicates a significant correlation between TTIs and average differential speeds during the P&D time intervals. However, outside of the

P&D time intervals, there is a noticeable decrease in speeds accompanied by a relative increase in TTIs in three of the cities, albeit with low correlation $R^2$ values. This underscores TTI as a reasonable indicator of traffic speed reduction or delays in non-recurring traffic scenarios. Furthermore, it underscores the impact of P&D movement on traffic performance, leading to pronounced congestion from the onset of the movement until its conclusion in the surrounding links.

### 4.2. TTIs and Differential Speeds

The average differential speed indicates a delay if it possesses a negative value. The averages of the TTIs and differential speeds were calculated in the P&D time intervals for time intervals across selected Saudi cities including Hassa, Makkah, Ha'il, Najran, Abha, and Riyadh, as illustrated in Table 3. This analysis provides valuable insights into the efficiency of traffic flow during crucial school pick-up and drop-off periods across diverse urban settings within the country. The results indicated that the TTI values during afternoon pick-up periods are higher than those during morning drop-off times, indicating a greater impact of disruptions and resulting congestion on links during pick-up intervals. This congestion and consequent delays stem from non-recurring movements. Riyadh exhibits the highest TTI values, attributable to its nature as the capital city, boasting a total population of 7.6 million inhabitants and 16 million daily traffic trips across its extensive 15,138 km road network. Other cities exhibit TTIs with values in close proximity to each other. Also, it can be noticed that the average TTI outside of the P&D time intervals is significantly higher compared to other values, suggesting adverse effects of the P&D movement on traffic performance. This can be attributed to the fact that as TTI values increase, the variability in congestion delay on trips also increases [30].

**Table 3.** The average TTI and differential speeds for P&D time intervals.

| City | Drop-Off (Morning) | | | | Pick-Up (Noon) | | | |
|------|------|------|------|------|------|------|------|------|
| | ΔSpeed | | ΔTTI | | ΔSpeed | | ΔTTI | |
| | μ | S.E. | μ | S.E. | μ | S.E. | μ | S.E. |
| Hassa | −5.87 | 0.39 | 1.18 | 0.04 | −5.93 | 0.46 | 1.19 | 0.02 |
| Makkah | −5.21 | 0.38 | 1.14 | 0.14 | −5.75 | 0.58 | 1.20 | 0.02 |
| Ha'il | −5.03 | 0.40 | 1.18 | 0.02 | −5.58 | 0.67 | 1.19 | 0.03 |
| Najran | −5.17 | 0.94 | **1.13** | 0.03 | −4.52 | 0.94 | **1.15** | 0.03 |
| Abha | −6.39 | 0.62 | 1.17 | 0.03 | −6.25 | 0.84 | 1.21 | 0.04 |
| Riyadh | −9.25 | 2.15 | **1.47** | 0.18 | −9.15 | 1.28 | **1.34** | 0.06 |

ΔSpeed: The average speed for each city; ΔTTI: The average of TTI for each city; S.E.: The standard error.

### 4.3. PTIs, BTIs, and TTIs for the Posted Speeds

The average values of the TTIs for each posted speed in links are estimated and summarized in Table 4. The mean travel times range between 1 and 4 min./mile, while the 95th% travel times range between 1 and 5 min./mile. PTIs range between 0 and 3, whereas BTIs range between 0 and 150. TTIs fall between 1.14 and 1.28.

**Table 4.** A synopsis of posted speed limit's performance metrics for travel time indices.

| Speed Limit (mph) | Average Travel Time (Minutes/Mile) | The 95th Percentile Travel Time (Minutes/Mile) | Planning Time Index (PTI) | Buffer Time Index (BTI) | Travel Time Index (TTI) |
|------|------|------|------|------|------|
| <40 | 3.89 | 4.14 | 1.22 | 24.31 | 1.28 |
| 40–55 | 1.68 | 2.21 | 1.34 | 28.27 | 1.18 |
| >55 | 1.12 | 1.71 | 1.51 | 34.26 | 1.14 |

### 4.4. TTIs with PTIs and the Posted Speed

For P&D movements, the interrelationship between TTIs and PTIs is estimated and depicted in Figure 5. Given the positive correlation between TTIs and PTIs, each TTI can be utilized to designate P&D movement in relation to the posted speed. The thresholds for each posted speed for the three indices were calculated. Previous studies indicate a non-linear interrelationship between BTIs and the posted speed. Accordingly, the relationship may exhibit an exponential increase followed by a decrease. Therefore, only the interrelationship between TTIs and PTIs is estimated to determine LOS [30].

For P&D movement, the interrelationship between TTIs and posted speed is shown in Figure 6. As posted speeds increase, TTIs decreased. Furthermore, the interrelationship between the 95th% travel times and posted speed is depicted in Figure 7.

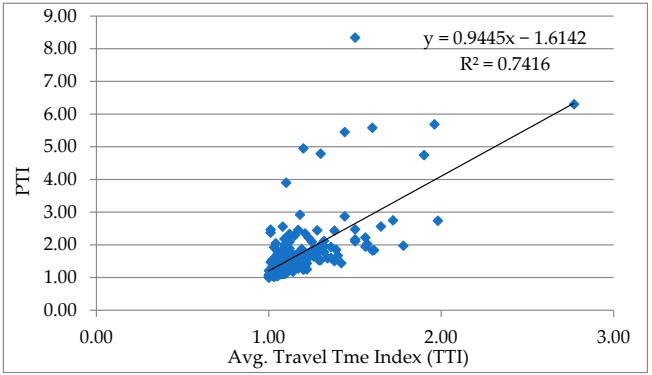

**Figure 5.** The average TTIs and PTIs interrelationship.

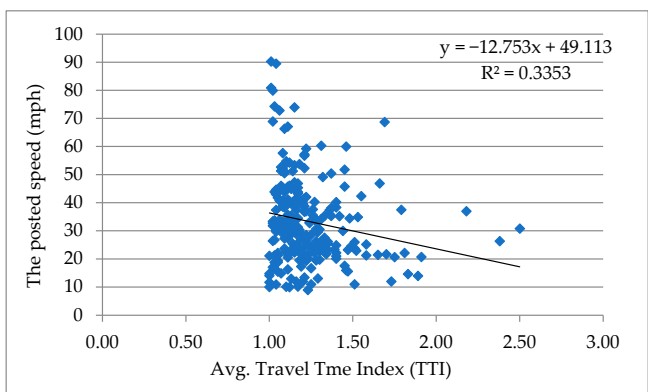

**Figure 6.** The average TTIs and the PSL (mph) interrelationship.

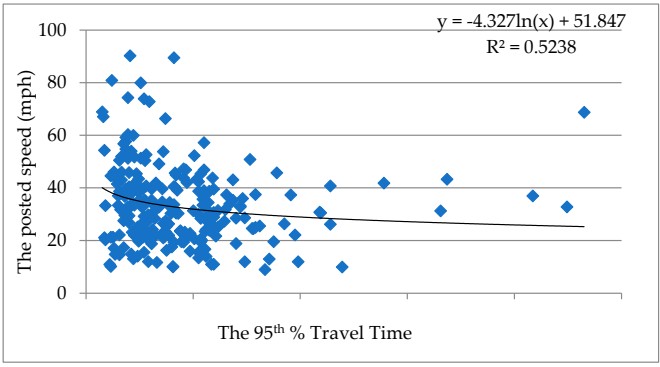

**Figure 7.** The PSL and the 95th% travel time (mph) interrelationship.

The previous scatter plots exhibit the best-fit trend model (equation) that was produced. In the 95th% travel times scatter logarithmic chart, the $R^2$ value emphasized their high consistency, standing at 0.5238, as shown in Figure 7. The trends show a corresponding increase in the 95th% travel times as the estimated speed decreases. Furthermore, the mean travel time to the 95th% travel time ratios are 1.46, 1.32, and 1.21 for posted speed limit categories with speeds of <40, 40–55, and ≥55 mph, respectively.

### 4.5. LOS Thresholds

For P&D time intervals, the speed range for each LOS category was substituted from the HCM into the best-fit trend line model to estimate its travel time measures-based LOS counterpart thresholds. These thresholds were estimated for each travel time interval, including the 95th% travel time, mean travel time, PTI, and average TTIs. Additionally, each speed mean travel time is included following the HCM. Here, the 95th% travel time in addition to the P&D movement indices are calculated using the estimated relationships. The estimated travel time measures-based LOS thresholds are compiled in Table 5.

**Table 5.** LOS thresholds—travel time-based performance measures in LOS for P&D time intervals.

| LOS | HCM Speed (mph) Thresholds | Mean Travel Time (Min/Mile) | 95th Percentile Travel Time (Min/Mile) | Planning Time Index (PTI) | Travel Time Index (TTI) |
|---|---|---|---|---|---|
| **<40 mph** | | | | | |
| A | 40 | ≤2.31 | ≤2.80 | 2.55 | 1.05 |
| B | 35 | 2.32–2.58 | 2.81–2.96 | 3.26 | 1.10 |
| C | 30 | 2.58–2.75 | 2.97–3.01 | 3.28 | 1.12 |
| D | 25 | 2.76–2.72 | 3.02–3.78 | 3.07 | 1.13 |
| E | 20 | 2.73–3.14 | 3.79–4.12 | 4.61 | 1.18 |
| F | <20 | >3.14 | >4.12 | 4.61 | 1.18 |
| **40–55 mph** | | | | | |
| A | 50 | ≤1.46 | ≤1.92 | 2.19 | 1.13 |
| B | 45 | 1.47–1.63 | 1.93–2.15 | 2.41 | 1.16 |
| C | 40 | 1.64–1.78 | 2.16–2.34 | 2. 62 | 1.17 |
| D | 35 | 1.79–1.88 | 2.35–2.48 | 2.80 | 1.19 |
| E | 30 | 1.89–1.77 | 2.48–2.56 | 2.89 | 1.22 |
| F | <25 | >1.78 | >2.56 | 2.89 | 1.22 |
| **>55 mph** | | | | | |
| A | 70 | ≤0.82 | ≤1.19 | 1.34 | 1.17 |
| B | 66 | 0.83–0.38 | 0.20–0.46 | 0.53 | 1.21 |
| C | 62 | 0.39–1.10 | 0.47–1.59 | 1.78 | 1.27 |
| D | 58 | 1.11–1.14 | 1.60–1.67 | 1.87 | 1.29 |
| E | 55 | 1.15–1.23 | 1.68–1.78 | 1.95 | 1.38 |
| F | <55 | >1.23 | >1.79 | 1.95 | 1.38 |

A similar pattern was exhibited in the interrelationship scatter plots of the mean travel time and the 95th% travel time thresholds as they surpassed the thresholds for average travel time. Nevertheless, the PTI and BTI thresholds grew and then decreased from (A to F) LOS, regardless of the PSL category. They are, however, typically low for either B or C LOS. The results, in conjunction with the polynomial distribution, explain the connection between the calculated speeds from the estimated models and both the PTIs and TTIs, suggesting that caution should be used when evaluating performance or predicting future needs utilizing the thresholds along with the measured values. This may clarify the fact of the regular accomplishment of the thresholds as a result of a congested link with high traversal times.

As seen, the mean travel time thresholds (measured in minutes/mile) were changed from (A to F) LOS. The 40–55 mph PSL groups have the highest mean travel time thresholds, while the >55 mph PSL group has the lowest. The mean travel time on links with high speed varies relatively slightly, ranging from <0.82 min/mile for LOS A to >1.23 min/mile for LOS F. But, in the case of the 30–40 mph PSL group, it varies from <1.46 min/mile for LOS A to >1.78 min/mile for LOS F. This is to be expected given the improved high-speed design standards.

In each PSL group, the thresholds for TTIs were changed from (A to F) LOS. The 40–55 mph PSL has the highest mean travel time threshold, while the PSL group > 55 mph has the lowest. TTIs on links with high speed vary relatively slightly, ranging from <1.17 min/mile for LOS A to >1.38 min/mile for LOS F. However, in the case of the PSL of 30–40 mph, it varies from <1.13 min/mile for LOS A to >1.22 min/mile for LOS F. In the P&D time intervals for both drop-off and pick-up, the estimated LOS of Riyadh, Hassa, Ha'il, Najran, Makkah, and Abha according to the relation between the average TTIs and PTIs are illustrated in Table 6.

**Table 6.** The LOS—travel time-based performance measures in each city for P&D time intervals.

| City | Avg. Speed Group | Drop-Off | | Pick-Up | |
|---|---|---|---|---|---|
| | | ΔTTIs | LOS | ΔTTIs | LOS |
| **Hassa** | 40–55 mph | 1.18 | C | 1.19 | D |
| **Makkah** | >55 mph | 1.14 | A | 1.20 | B |
| **Ha'il** | 40–55 mph | 1.18 | C | 1.19 | D |
| **Najran** | <40 mph | 1.13 | D | 1.15 | D |
| **Abha** | 40–55 mph | 1.17 | C | 1.21 | E |
| **Riyadh** | >55 mph | 1.47 | F | 1.34 | E |

For Makkah, the average LOS is the best at P&D time intervals is between A and B in the afternoon and morning, respectively, is the worst LOS is illustrated for P&D time intervals out of those intervals. The city is most affected by other factors such as commercial activities and the geometric characteristics of links, which cause congestion all the time during the daytime besides the P&D movements. For the rest of the cities, the LOS ranges between C and D. The cutting-edge strategies for congestion due to P&D will help in highlighting its impact.

## 5. Recommended Policies for Congestion-Free Pick-Up and Drop-Off

By implementing such innovative policies and strategies, schools can effectively reduce congestion during P&D times while promoting sustainable and safe transportation options for students and families. Collaboration between schools, local authorities, transportation agencies, and the community is essential to successfully implement and sustain these initiatives as presented in Table 7. Understanding the dynamics of pick-up and drop-off traffic enables authorities to develop strategies seeking to alleviate congestion and mitigate safety risks in school zones. These efforts contribute to the creation of more livable and sustainable communities while ensuring the safety and well-being of their young residents.

**Table 7.** Innovative policies and strategies to alleviate congestion and mitigate safety risks in school zones.

| Recommended Policies | Methodology | Outcomes |
|---|---|---|
| **Staggered Timings** | - Using different timing of students' P&D. | Optimizing traffic strategies. |
| **Designated Zones** | - Designate specific drop-off/pick-up zones and/or road markings to guide drivers. | Improving traffic safety. |

**Table 7.** *Cont.*

| Recommended Policies | Methodology | Outcomes |
|---|---|---|
| **Car sharing Initiatives** | - Facilitating car sharing among neighboring families. | Improving traffic performance. |
| **P&D Loop Efficiency** | - Loop design optimization for P&D to maximize efficiency.<br>- A one-way traffic flow with separate lanes for students dropping off and picking up;<br>- Local government personnel for managing traffic smooth flow. | Improving traffic performance. |
| **Distant P&D stations** | - Identify neighborhood sites for students' P&D.<br>- Busing or walking to schools. | Improving traffic performance. |
| **School Bus route scheduling** | - School bus scheduling (route and timing) to facilitate students' safety and comfort.<br>- Use technological means to track buses' trips in real-time. | Optimizing traffic strategies. |
| **Flexible P&D sites** | - Allow parents to choose from multiple P&D sites based on their convenience. | Optimizing traffic strategies. |
| **Periodical societal Evaluation** | - Periodical checks of the effectiveness of school P&D policies via societal surveys. | Optimizing traffic strategies. |

## 6. Conclusions

Researchers, city planners, and legislators are increasingly interested in comprehending, identifying, and addressing the prominent aspects that influence daily occurrences in order to deploy potential solutions to improve traffic safety in this delicate area. This interest arises from the dynamic interaction between school students' P&D movements and traffic operational performance. As a realistic case study, real-life traffic datasets of adjacent roads around 40 schools in six different cities across the KSA were comprehensively analyzed. Using the Google Maps API, real-time interval data for 242 links around schools were collected.

This study aims to analyze the traffic operational performance by calculating TTIs by estimating the travel time to free flow conditions travel time ratio for each road segment during three main selected time intervals, namely drop-off (morning), pick-up (afternoon), and off-peak periods. Additionally, the HCM method is employed for a comprehensive assessment of LOS based on a wide range of factors such as PTI and TTI for non-recurring movements (e.g., school pick-up and drop-off periods, special events, accidents...etc.) can be highly important. The average LOS thresholds are estimated for PSL of related links surrounding schools during P&D time intervals and outside of those intervals. The study concluded the following:

- There is a negative correlation between the average TTIs and the PSL during P&D time intervals.
- Riyadh is the city most affected by school P&D movements and traffic despite the high road speeds. This is attributed to factors such as commercial activities and the geometric characteristics of links, leading to congestion throughout the daytime, in addition to P&D movements.
- For the rest of the cities, LOS ranges between C and D.

By grasping the intricacies of pick-up and drop-off traffic patterns, policymakers can draw up strategies to mitigate congestion and enhance the safety conditions in school zones. This proactive approach fosters the development of communities that are both sustainable and conducive to raising the quality of life and safeguarding welfare. It can be noticed that speeds decrease as TTIs increase, with the highest value observed during pick-up time intervals compared to drop-off time intervals, albeit with relatively increasing TTIs in both scenarios across all cities.

Other factors such as commercial activities and the geometric characteristics of surrounding roads can further enhance the operational performance. Future research endeavors could focus on integrating these variables with other spatial data to refine the temporal-spatial model. This may include other influenced factors, i.e., the number of lanes, availability of parking facilities, presence of medians, implementation of speed calming and safety measures at school entrances, and sidewalk width around schools. Such comprehensive analyses would provide deeper insights into traffic dynamics and aid in the development of more effective strategies to improve traffic flow and safety around school zones.

**Author Contributions:** Conceptualization, S.S., A.A. and M.E.; methodology, S.S. and M.E.; software, S.S. and M.E.; validation, S.S.; analysis, S.S. and M.E.; data S.S. and A.A.; writing—original draft preparation, S.S. and M.E.; writing—review and editing, S.S., A.A. and M.E. All authors have read and agreed to the published version of the manuscript.

**Funding:** This research received no external funding. The APC was funded by the authors.

**Institutional Review Board Statement:** Not applicable.

**Informed Consent Statement:** Not applicable.

**Data Availability Statement:** The used data are unavailable due to privacy issues.

**Acknowledgments:** The authors express their gratitude to The Center of Road Traffic Safety at Naif Arab University for Security Sciences for providing the facilities required for completing this research.

**Conflicts of Interest:** The authors declare that they have no financial/personal interests or any other matters that could affect the objectivity this paper. The authors certify that potential competing interests do not exist for this paper.

## Abbreviations

| | |
|---|---|
| BTI: | Buffer Time Index |
| CPM: | Congestion Performance Measure |
| HCM: | Highway Capacity Manual |
| LOS: | Level of Service |
| P&D: | Pick-up and Drop-off |
| PSL: | Posted Speed Limits |
| PTI: | Planning Time Index |
| TTI: | Travel Time Index |
| TTR: | Travel time reliability |

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
