# Peer review of "Analyzing the Traffic Operational Performance of School Pick-Up and Drop-Off Dynamics in Saudi Arabia"

_sustainability, doi:10.3390/su16125154_

Round 1
Reviewer 1 Report
Comments and Suggestions for Authors
This study focuses on the inter-relationship between school students' Pick-up and Drop-off (P&D) movements and traffic operational performance. By using Google Map API to calculate the travel time around 40 schools in six cities in Saudi Arabia during peak school hours, further exploring the relationship between traffic performance indicators such as differential speeds, Travel Time Index (TTI), and thresholds of Level of Service (LOS). However, some minor changes in the manuscript should be modified. Therefore, I recommend reconsideration of the manuscript following major revision.
1. In the section 4, it is suggest to provide more details on how the 40 schools were selected, the criteria for school selection, and the specific data points collected from the real-world traffic data set. This will help establish the validity and reliability of the findings.
2. In the section 4, the manuscript mentions using Python scripts and Google API to estimate travel time, but lacks specific description and validation of these methods. For example, there is no detailed explanation of the algorithm design, parameter settings, and accuracy of API calls in Python scripts.
3. In the section 5, the manuscript mentions the trend of changes in traffic performance indicators (such as PTI, TTI, and LOS), but don’t conduct in-depth analysis of the results. For example, there is no exploration of key factors that affect traffic performance (such as school size, road network structure, traffic management strategies, etc.), nor is there a detailed comparison and explanation of differences between different cities.
4. In the section 7, the manuscript mentions that the research findings provide valuable insights for urban planners and policymakers, but does not explicitly propose specific policy recommendations. Based on the research results, further exploration can be conducted on how to improve the traffic performance and safety conditions around schools by improving road design, optimizing traffic management strategies, and encouraging public transportation.
Author Response
The authors would like to thank the reviewer for his comments and positive feedback on our manuscript. We have revised our paper following these comments and we do believe that they have further improved the paper. Please find the following our detailed responses.

Reviewer 2 Report
Comments and Suggestions for Authors
The paper presents an interesting reserch topic but before publication it is important to correct some parts of the paper in order to make the explanation of the conducted reserch and results more clear.
- The structure should be corrected - introduction should be divided from the state-of-the-art part as this second part is very long. In these two parts everything should be supported with referencies, text should be without parts that are expleined few times (e.g. the traffic problems in front of the schools should be described just ones in the text)
- all the abbreviations should be explained (some are but some are not so it is difficult to go through the text)
- Methodology should be presented graphically and part 2. of the paper included in the explanation of methodology or better to say - Materials and Methods chapter. Data collection is also part of Materials and Methods chapter.
- Every step of the methodology should be described - road chategories and their atributes should be presented in the table in which also data from Figure 1 can be added. In this way readers will have more information about experimental part of the research.
- Analyses and Results - it is important to say at the beginning of the chapter what will be analysed
- when figures are mentioned in the text it should be clear to which figures authors refer (e.g. line265)
- TTIs should be explained better - for which conditions are defined?
- the conclusion (line 280-281) is expected, analyses should focus on how and in which circumstances less/more
- I would suggest to use SI units
- I would suggest to devide Discussion and Conclusions from more "practical" part of the paper - suggestions of possible solutions to mitigate analyzed problem of congestion
Comments on the Quality of English Language
English language editing is needed.
Author Response
On behalf of the authors, I would like to express my sincere gratitude upon receiving your reviewing comments. The authors are very thankful and indebted to the editor and reviewers for their profound and thorough review and the positive assessment of our manuscript “sustainability-3010192”. The manuscript has been revised accordingly considering the reviewers’ useful suggestions and highly appreciated comments. The authors do believe that the paper has been further improved upon such fruitful insights. The authors hope that the latest revision has improved the paper to the level of reviewers’ satisfaction. Please find the following detailed responses.

Reviewer 3 Report
Comments and Suggestions for Authors
The paper studies the mobility behavior in the environment of 40 educational centers in 6 different cities in the Kingdom of Saudi Arabia, specifically during the class start and end times, based on planning time indexes (PTI) and travel time indexes (TTI). It is an interesting article because, as mentioned by the authors, the new tools offered by companies such as Google can be useful for decision making in urban planning. In the results obtained
In order to improve the quality of the article, several changes should be made:
1. Provide complete information on data collection so that the experiments are reproducible. Or provide a link in the cloud to the data in GeoJSON format or something similar.
2. How were the 40 schools used in the study selected?
3. In the conclusions, regarding the city of Riyadh, it should be justified that this is the city most affected by factors such as commercial activities and geometric characteristics of the links. What happens when a map is created integrating the results obtained with the geographical locations of highly frequented commercial locations?
4. In the proposed methodology, is it possible to integrate information on accidents, crashes or weather conditions?
5. The unit of measurement is omitted in most of the figures. For example, in Figure 4, is the TTI expressed in minutes or in hours?
Author Response

(The authors gave the same response as above.)

Reviewer 4 Report
Comments and Suggestions for Authors
Dear Authors,
thank you for your research but please look at the comments:
1. In the second chapter, materials and methods should include a methodology figure/scheme. Explained to us (readers) how the whole process was obtained.
2. For improved readability and clarity, consider including an abbreviation table listing all defined terms used within the manuscript.
3. In all figures for consistency, please ensure the font matches the Palatino Linotype typeface and size used in the main body of the text.
4. All equations should be referenced in the text.
Thank you
Author Response

(The authors gave the same response as above.)

Round 2
Reviewer 3 Report
Comments and Suggestions for Authors
The authors have improved the paper and addressed the significant required changes in the first version of this paper. The paper is ready for publication.